# Susceptibility of Sea Bream (*Sparus aurata*) to AIP56, an AB-Type Toxin Secreted by *Photobacterium damselae* subsp. *piscicida*

**DOI:** 10.3390/toxins14020119

**Published:** 2022-02-05

**Authors:** Inês Lua Freitas, Alexandra Teixeira, Inês Loureiro, Johnny Lisboa, Aurélia Saraiva, Nuno Miguel Simões dos Santos, Ana do Vale

**Affiliations:** 1Fish Immunology and Vaccinology Group, IBMC—Instituto de Biologia Molecular e Celular, Universidade do Porto, 4200-135 Porto, Portugal; ines.freitas@ibmc.up.pt (I.L.F.); alexandrat@ibmc.up.pt (A.T.); ines.loureiro@ibmc.up.pt (I.L.); johnny.lisboa@ibmc.up.pt (J.L.); 2Fish Immunology and Vaccinology Group, i3S—Instituto de Investigação e Inovação em Saúde, Universidade do Porto, 4200-135 Porto, Portugal; 3Biology Department, Faculty of Sciences, University of Porto, 4169-007 Porto, Portugal; amsaraiv@fc.up.pt; 4CIIMAR—Interdisciplinary Center of Marine and Environmental Research of the University of Porto, 4450-208 Matosinhos, Portugal

**Keywords:** photobacteriosis, pseudotuberculosis, fish pasteurellosis, *Sparus aurata*, *Photobacterium damselae* subsp. *piscicida*, AIP56, AB toxin, virulence factor

## Abstract

*Photobacterium damselae* subsp. *piscicida* (*Phdp*) is a Gram-negative bacterium that infects a large number of marine fish species in Europe, Asia, and America, both in aquacultures and in the natural environment. Among the affected hosts are economically important cultured fish, such as sea bream (*Sparus aurata*), sea bass (*Dicentrarchus labrax*), yellowtail (*Seriola quinqueradiata*), and cobia (*Rachycentron canadum*). The best characterized virulence factor of *Phdp* is the Apoptosis-Inducing Protein of 56 kDa (AIP56), a secreted AB-type toxin that has been shown to induce apoptosis of sea bass phagocytes during infection. AIP56 has an A subunit that displays metalloprotease activity against NF-kB p65 and a B subunit that mediates binding and internalization of the A subunit in susceptible cells. Despite the fact that the *aip56* gene is highly prevalent in *Phdp* isolates from different fish species, the toxicity of AIP56 has only been studied in sea bass. In the present study, the toxicity of AIP56 for sea bream was evaluated. Ex vivo assays showed that sea bream phagocytes are resistant to AIP56 cytotoxicity and that resistance was associated with an inefficient internalization of the toxin by those cells. Accordingly, in vivo intoxication assays revealed that sea bream is much more resistant to AIP56-induced lethality than sea bass. These findings, showing that the effect of AIP56 is different in sea bass and sea bream, set the basis for future studies to characterize the effects of AIP56 and to fully elucidate its virulence role in different *Phdp* susceptible hosts.

## 1. Introduction

AIP56 (Apoptosis-Inducing Protein of 56 kDa) is an AB toxin secreted by virulent strains of *Photobacterium damselae* subsp. *piscicida* (*Phdp*) [1], a Gram-negative bacterium that causes a septicemic infection that leads to high mortalities in a large number of wild and cultured warm water marine fish species in Europe, Asia, and North America [2]. *Phdp* is a ubiquitously distributed pathogen resistant to many antimicrobials that constitutes a serious threat for the aquaculture production of economically important species, such as sea bream (*Sparus aurata*) [3], sea bass (*Dicentrarchus labrax*) [4], sole (*Solea senegalensis* and *Solea solea*) [5], yellowtail (*Seriola quinqueradiata*) [6], and cobia (*Rachycentron canadum*) [7].

*Phdp* infections are characterized by the occurrence of generalized bacteremia and extensive cytopathology with abundant tissue necrosis [8]. Whitish tubercle-like lesions of about 0.5 to 3.5 mm in diameter, consisting of accumulations of bacteria, apoptotic cells, and necrotic cell debris are often present in several internal organs of infected fish [8,9,10,11,12]. Many of the *Phdp*-induced pathological changes have been attributed to AIP56, an exotoxin secreted in high amounts by the T2SS of *Phdp* [1,13]. Studies in sea bass have shown that AIP56 is secreted and systemically disseminated during infection and causes apoptosis of macrophages and neutrophils, ultimately leading to their extensive lysis by post-apoptotic secondary necrosis [1,8,14]. This disarms the host phagocytic defense and allows the massive extracellular proliferation of the pathogen, thereby contributing to establishment of the infection [8]. Additionally, the massive lysis of host phagocytes induced by the toxin results in the release of highly cytotoxic molecules, including neutrophil elastase, a situation that induces tissue damage and contributes to the genesis of the necrotic lesions characteristic of *Phdp* infections [8,14]. Previous ex vivo and in vitro studies have shown that AIP56 is a short-trip AB toxin that displays toxicity towards sea bass and mouse macrophages [15,16]. AIP56 toxicity involves the activity of its zinc-metalloprotease domain that catalyzes the proteolytic cleavage of NF-kB p65 in susceptible cells, leading to depletion of this transcription factor and, ultimately, to the cell’s apoptosis [1,16]. To reach the cytosol, where its substrate is located, AIP56 follows a multi-step process that starts by the binding to a so far unknown receptor at the surface of the target cell, followed by receptor mediated endocytosis of the toxin [15]. Translocation into the cytosol has been proposed to occur from early/recycling endosomes, by a mechanism that requires endosome acidification [15] and the participation of host cell Hsp90 [17].

Although *aip56* is almost 100% prevalent in *Phdp* isolated from disease outbreaks in different fish species over last three decades [18]; the effects of AIP56 have only been studied in sea bass. In this study, the toxicity of AIP56 for sea bream, another economically important fish species susceptible to *Phdp* infections, was investigated. Ex vivo assays showed that sea bream phagocytes are resistant to AIP56-induced cytotoxicity, which is in agreement with the results obtained in vivo showing that sea bream is highly resistant to AIP56-induced lethality. The here reported differences in the susceptibility to AIP56 between sea bass and sea bream emphasize the need for additional studies to determine the susceptibility to the toxin and its role for virulence in other *Phdp* susceptible hosts. The clarification of the host-specific effects of the toxin will allow understanding the relative contribution of AIP56 for the establishment of *Phdp* infections in different hosts, which will likely impact the design of preventive and therapeutic measures against this bacterial disease.

## 2. Results

### 2.1. aip56 Is Ubiquitous among Sea Bream and Sea Bass Phdp Isolates

A previous study [18] analyzed 103 *Phdp* strains isolated between 1963 and 2015 from different host species and geographical locations, revealing an almost 100% prevalence of the *aip56* gene, which was only absent in six of the isolates, two of them (ATCC29690 and EPOY8803-II) previously shown to be non-virulent [19]. Here, the analysis of the distribution of *aip56* was extended to 49 *Phdp* field isolates recovered from sea bream or sea bass during outbreaks occurred between 2014 and 2020 in different European countries (Table 1, rows 5–53). As shown in Figure 1, all these isolates tested positive for *aip56*.

### 2.2. Sea Bream and Sea Bass Phdp Isolates Secrete Identical AIP56 Toxins

To evaluate if sea bream and sea bass isolates secrete identical AIP56 toxins, the *aip56* ORF from 3 sea bream (DI21, GRSA19-1 and SPSA20-2) and 3 sea bass (MT1415, GRDL19-1 and SPDL20-5) strains isolated in different geographical locations were sequenced and the nucleotide and deduced amino-acid sequences compared. The results revealed that the *aip56* ORF from those strains are 100% identical, indicating that they secrete identical AIP56 toxins.

### 2.3. AIP56 Does Not Induce Apoptosis or Other Signs of Toxicity in Sea Bream Peritoneal Leukocytes Ex Vivo

The susceptibility of sea bream leukocytes to AIP56 was tested using peritoneal leukocytes collected from resting peritoneal cavities or from peritoneal cavities that were stimulated with Incomplete Freunds’ Adjuvant (IFA) 12 days prior to leukocytes’ collection. The peritoneal leukocyte populations from resting sea bream consisted mainly of neutrophilic and acidophilic granulocytes (47.96 ± 0.09% and 44.15 ± 0.09%, respectively), with small numbers of macrophages (4.10 ± 0.03%) and lymphocytes (3.7 ± 0.01%), whereas leukocyte populations collected from inflamed peritoneal cavities were composed mainly of monocytes/macrophages (88.09 ± 0.04%), with small numbers of neutrophils (1.20 ± 0.01%), acidophilic granulocytes (5.77 ± 0.02%), and lymphocytes (4.93 ± 0.03%). To test the susceptibility of the cells to AIP56, they were incubated with the toxin for 5 h and the occurrence of apoptosis or other signs of toxicity was evaluated by analyzing cytospin preparations by light microscopy (Figure 2). Sea bass leukocytes collected from inflamed peritoneal cavities, mainly composed of macrophages (82.6 ± 0.08%) and containing small numbers of neutrophils (7.64 ± 0.05%), eosinophilic granular cells (2.10 ± 0.01%), and lymphocytes (7.40 ± 0.03%) were used as a positive control, because it is well established that sea bass macrophages are highly susceptible to AIP56 and undergo apoptosis in response to intoxication [8,20]. As shown in Figure 2, incubation of resting or inflamed sea bream peritoneal leukocytes with AIP56 did not lead to an increase in the percentage of cells with apoptotic morphology or with any other morphological alterations indicative of toxicity, when compared to untreated cells. In contrast, and as expected, most sea bass leukocytes were apoptotic after incubation with AIP56 (Figure 2), confirming that the toxin was active. These results suggest that sea bream peritoneal leukocytes are resistant to AIP56 intoxication.

### 2.4. AIP56 Displays Proteolytic Activity against Sea Bream NF-kB p65

Previous studies have shown that in susceptible cells (e.g., sea bass peritoneal macrophages and mouse macrophages), AIP56 cleaves NF-kB p65 at a conserved Cys–Glu peptide bond located at the REL homology domain, leading to depletion of this transcription factor and apoptotic death of the cells [1,16]. A possible explanation for the lack of toxicity of AIP56 for sea bream leukocytes could be the inability of the toxin to cleave NF-kB p65 of this fish species. To investigate this possibility, we performed a multiple alignment of the translated amino-acid sequences of NF-kB p65 REL homology domain from different species (Figure 3A), which showed that AIP56 cleavage site is conserved in sea bream NF-kB p65, suggesting that this protein is also susceptible to AIP56 cleavage. To confirm this suggestion, ^35^S-labeled sea bream NF-kB p65 was incubated in vitro with AIP56, and cleavage accessed by autoradiography. ^35^S-labeled sea bass NF-kB p65 was used as a positive control [16]. As shown in Figure 3B, AIP56 cleaves sea bream and sea bass NF-kB p65 with similar efficiency. As expected, no cleavage occurred after incubation with the catalytically inactive AIP56^AAIVAA^ [16]. These results clearly show that sea bream NF-kB p65 is susceptible to proteolytic cleavage by AIP56.

### 2.5. The Internalization of AIP56-488 in Sea Bream Peritoneal Leukocytes Is Inefficient

Studies performed in sea bass and mouse macrophages have shown that intoxication by AIP56 is a multi-step process that begins with clathrin-dependent endocytosis, followed by low pH-dependent translocation of the toxin from the endosomes into the cytosol, where it cleaves NF-kB p65 [15]. Since the resistance of sea bream leukocytes to AIP56 toxicity cannot be explained by an inability to cleave sea bream NF-kB p65, we decided to investigate if it could be related to an incapacity of the toxin to enter sea bream cells. For this, peritoneal leukocytes collected from resting or inflamed peritoneal cavities of sea bream were incubated with fluorescent-labeled AIP56 (AIP56-488) for 15 min on ice, followed by 15 min at 22 °C and the internalization of the toxin was analyzed by flow cytometry. Peritoneal leukocytes collected from IFA-stimulated sea bass, which are known to internalize AIP56 [15], were used as a positive control. In these experiments, we used cells from the same batches of fish used for collecting cells for the in vitro toxicity tests (Figure 2), i.e., sea bream Batch IV (75–158 g) and sea bass Batch II (202–367 g), but also cells collected from sea bream Batch II (6.6–14.8 g) or sea bass Batch I (43–65 g). As shown in Figure 4A,B, almost no AIP56-488-positive cells were detected in sea bream leukocytes collected from resting peritoneal cavities, which may explain the observed resistance of those cells to AIP56-induced cytotoxicity (Figure 2). Additionally, only a minority of sea bream leukocytes collected from IFA-stimulated sea bream (17.7% and 29.7% for Batch II and IV, respectively) became positive for AIP56-488 (Figure 4A,B), indicating that most sea bream macrophages that are recruited to the peritoneal cavity in response to IFA stimulation are unable to internalize the toxin. This contrasts with the results obtained in IFA-stimulated sea bass peritoneal leukocytes, in which toxin internalization was detected in more than 70% of the cells (75.9% and 74.2% for Batch I and II, respectively). Moreover, the intensity of fluorescence in sea bass cells positive for AIP56-488 was higher than in AIP56-488-positive cells from sea bream (Figure 4C), suggesting that the toxin binds and is internalized more efficiently by sea bass macrophages. The lower efficiency of AIP56 internalization in sea bream macrophages may contribute to the fact that even though the toxin is internalized by some sea bream macrophages, no AIP56-induced toxicity was detected in these cells.

### 2.6. Sea Bream Are Highly Resistant to AIP56 Toxicity

To investigate the susceptibility of gilthead sea bream to AIP56, six independent experiments were performed (Table 2). In each experiment, groups of sea bream were injected intra-peritoneally (i.p.) with different doses of AIP56 followed by lethality assessment. Groups of sea bass, which are known to be susceptible to AIP56 [8], were injected in parallel to confirm that the AIP56 was active. In all experiments, the injection of AIP56 in sea bass resulted in mortality. In this species, toxin doses ranging from 0.3–1.0 μg/g body weight resulted in 100% mortality and 60–90% mortality was observed for doses ranging from 0.005–0.2 μg/g body weight. In contrast, no sea bream mortality was observed in experiments 1, 2, 3, 5, and 6, after i.p. injection of AIP56 in doses ranging from 0.9–4.2 μg/g body weight, indicating that this fish species is highly resistant to AIP56 toxicity. Sea bream mortality was however observed in experiment 4, in which 2.2 or 0.2 μg AIP56/g body weight resulted in 100% and 30% mortality, respectively. Nevertheless, despite the increased susceptibility of this batch of sea breams to AIP56, the results of experiment 4 confirmed the higher resistance of sea bream to intoxication, when compared to sea bass, as 90% mortality was observed after injection of sea bass with 0.2 μg AIP56/g body weight, compared to the 30% mortality obtained in sea bream with the same dose. The observed sea bream mortality was caused by the toxin and involved its catalytic activity, as no mortality was observed after injection of the vehicle or 2.2 μg/g body weight of the catalytically inactive AIP56^AAIVAA^.

The high resistance of sea bream to AIP56 intoxication suggested that the pathological effects of the toxin for this fish species were mild or even absent. To assess this, spleen, head-kidney and liver samples of fish injected with AIP56 (or vehicle, as control) were collected for histological examination. The first batch of samples were from experiment 3 (Table 2), and included samples from sea bream injected with 3.1 μg AIP56/g body weight, in which no mortality was observed, from sea bass injected with 0.7 μg AIP56/g body weight, that led to 100% mortality, and vehicle controls. The second batch of samples were collected in experiment 4 and included samples from sea bream injected with 2.2 μg AIP56/g body weight, in which 100% mortality was observed, and vehicle controls. Organs collected from sea bass and sea bream injected with vehicle presented a typical histological structure (Figure 5, micrographs 1, 3 and 5; Figure 6, micrographs 1, 4 and 7). Hepatic steatosis, a common phenomenon in farmed fish, was evident in both species (Figure 5, micrographs 5; Figure 6, micrograph 7). In agreement with previous studies [8], i.p. inoculation of AIP56 in sea bass resulted in major histopathological changes in the spleen, head-kidney, and liver (Figure 5). Changes in the spleen included disorganization of the parenchyma architecture, red pulp atrophy, and multifocal necrosis (Figure 5, micrograph 2). A marked disorganization of parenchyma’s structure was also evident in the head-kidneys of AIP56-injected sea bass. In this organ, multifocal necrosis of the hematopoietic tissue was evident at 36 h post-injection (Figure 5, micrograph 4) and generalized necrosis was observed in moribund fish. Injection of AIP56 in sea bass resulted also in severe alterations in the liver, including loss of hepatic structure with occurrence of massive hepatocellular necrosis (Figure 5, micrograph 6) and pancreatic atrophy.

In comparison, and in agreement with the lethality results, the pathological alterations induced by AIP56 in sea bream were less prominent (Figure 6). Spleens from AIP56-injected sea breams presented a normal histological structure (Figure 6, panels 2 and 3). In head-kidneys collected from AIP56-injected sea breams from experiment 3, histological alterations, including the presence of degenerated cells and occurrence of multifocal necrosis, were observed in 4 out of 9 specimens (one collected at 36 h and three collected at 60 h post-injection). In experiment 4, pathological alterations were observed in all head-kidneys from AIP56-injected sea breams and were more exacerbated, with higher number of degenerated cells and more frequent and extensive necrotic foci (Figure 6, micrographs 5 and 6). Livers from sea bream injected with AIP56 in experiment 3 presented a normal histological structure, whereas degenerated cells and spotty/focal necrosis (Figure 6, micrographs 8 and 9) were present in all livers from AIP56-injected sea bream from experiment 4. Overall, these observations confirm that the pathology induced by AIP56 in sea bream is milder than in sea bass, while showing that the occurrence of sea bream mortality in response to toxin injection in experiment 4 was associated with histopathological changes in the head-kidney and liver.

## 3. Discussion

AIP56 is a plasmid encoded exotoxin abundantly secreted by virulent strains of *Phdp* in vitro and in vivo [1,8,13]. It has been shown that during *Phdp* infections in sea bass, AIP56 is systemically distributed and plays a major virulence role, by triggering the destruction of the sea bass professional phagocytes, macrophages and neutrophils [23], by post-apoptotic secondary necrosis, a process that culminates in massive phagocytes’ lysis [8]. This not only leads to impairment of the host phagocytic defense and, consequently, to unrestricted extracellular multiplication of *Phdp*, but also to the exposure of the host tissues to the highly cytotoxic molecules released from lysing phagocytes that likely participate in the genesis of the necrotic lesions typically associated to *Phdp* infections. Although a similar histopathology has been reported in advanced *Phdp* infections of different fish species, including striped bass (*Morone saxatilis*), sea bream, sea bass, and sole [8,10,24], to date, the contribution of AIP56 for the establishment of the infection in other *Phdp* hosts, apart from sea bass, has not been characterized.

In this work, the toxic effects of AIP56 for sea bream were investigated. We started by analyzing the susceptibility of the sea bream professional phagocytes—acidophilic granulocytes, which are the functional equivalents to mammalian neutrophils, and macrophages [25,26,27]—to AIP56 ex vivo. For this, we used peritoneal leukocytes collected from undisturbed peritoneal cavities, composed mainly by neutrophilic and eosinophilic granulocytes (approximately 48 and 44%, respectively) or leukocytes collected 12 days after i.p. injection of IFA, that were highly enriched in monocytes/macrophages (88% of the total population). Both leukocyte populations were found to be resistant to AIP56 toxicity. The resistance of the resting peritoneal leukocytes (e.g., neutrophilic and eosinophilic granulocytes) agrees with the results obtained in the toxin internalization assays, which showed that AIP56 is not internalized by those cells. For these studies, we used an AIP56 carrying a 6-histidine tag at the C-terminus, but it is unlikely that this could explain the lack of toxicity of AIP56 for sea bream cells, because previous studies have shown that his-tagged AIP56 remains toxic for sea bass [15,16], mouse [15] and human macrophages (unpublished results). The most likely explanation for the inability of sea bream neutrophilic and eosinophilic granulocytes to internalize the toxin is the lack of expression of the AIP56 receptor in those cells, since previous studies in sea bass and mouse macrophages revealed that the entry of AIP56 in the target cells is initiated by binding to a cell surface receptor that remains unidentified [15]. In agreement with this, in the present study, the percentage of sea bass IFA-stimulated peritoneal leukocytes that internalized the AIP56-488 (approximately 76 and 74% for Batch I and Batch II, respectively) was similar to the percentage of cells that underwent apoptosis in response to intoxication (approximately 74% for Batch II). In what concerns IFA-stimulated peritoneal leukocytes from sea bream, there was no concordance between toxin internalization and cytotoxicity, as they were found to be resistant to AIP56-induced toxicity, despite the fact that internalization of AIP56-488 was detected in a significant percentage of cells (approximately 18% and 30% for Batch II and Batch IV, respectively). There are two main hypotheses that can contribute to this discrepancy. On the one hand, the lower amounts of AIP56 internalized in sea bream macrophages, when compared to the sea bass counterparts, may not be sufficient to decrease the NF-kB p65 levels. On the other hand, since it is known that the consequences of NF-kB p65 downregulation in different cells may differ [28], it cannot be excluded that even if AIP56 is able to lower the levels of NF-kB p65 in some sea bream macrophages, they do not die in response to intoxication. In this work, although it was shown that AIP56 cleaves sea bass and sea bream NF-kB p65 with similar efficiency in vitro, it was not possible to determine the extent of p65 cleavage in AIP56-treated sea bream macrophages, due to the lack of antibodies able to recognize sea bream NF-kB p65. This is an important aspect that must be addressed in the future. Additionally, it would also be important to analyze the presence (e.g, by immunohistochemistry) of AIP56 in phagocytes from sea bream naturally infected with *Phdp*, to confirm the here reported in vitro observations in a more biologically relevant scenario.

In agreement with the observed low susceptibility of sea bream leukocytes to AIP56-induced toxicity, in vivo intoxication assays, performed with different batches of fish, revealed that sea bream are much less susceptible to AIP56-induced lethality than sea bass. In sea bass, mortality was observed in 6 out of 6 experiments, with toxin doses ranging from 0.005–1.0 μg/g body weight resulting in 60–100% mortality, whereas in sea bream, mortality occurred only in 1 out of 6 experiments. In this experiment, injection of sea bream with 0.2 or 2.2 μg AIP56/g body weight resulted in 30% and 100% mortality, respectively, whereas the mortality induced in sea bass by the lowest dose (0.2 μg AIP56/g body weight) reached 90%. Accordingly, the histopathological alterations induced by the toxin in sea bream were milder than in sea bass. The most striking difference was observed in the spleen. In sea bass, marked pathological changes were detected in this organ after injection of 0.7 μg AIP56/g body weight, a dose that led to 100% lethality. In contrast, no histological changes were observed in sea bream spleens even after injection of toxin dose that also led to 100% mortality (2.2 μg AIP56/g body weight). In line with these findings, the histological alterations detected in the head-kidneys and livers of sea bream injected with 2.2 μg AIP56/g body weight were less prominent than the alterations detected in sea bass injected with 0.7 μg AIP56/g body weight. The molecular basis of the high resistance of sea bream to AIP56 is currently unknown, but the results of the ex vivo experiments suggest that it may be related to differences in the expression levels of the AIP56 receptor in sea bream cells, both at the level of cell populations and individual cells, and/or to the existence of evolutionary differences between the sea bass and sea bream receptors that result in different binding affinities to AIP56. The elucidation of these possibilities is worth pursuing, but will require the identification of the AIP56 receptor.

Taken together, the results of the ex vivo and in vivo experiments indicate that sea bream is more resistant to AIP56 intoxication than sea bass, which may suggest that the toxin plays a less important virulence role during *Phdp* infections in sea bream. However, the screening of 49 *Phdp* isolates recovered between 2014 and 2020 from diseased sea bream and sea bass in different European countries showed a 100% prevalence of the AIP56-encoding gene. This result, which is in agreement with previous studies reporting a ubiquitous distribution of the *aip56* gene in *Phdp* strains recovered worldwide from different fish species in the last 30 years [18,29,30], suggests the existence of selective pressure to maintain the toxin gene and supports the hypothesis that in the majority of hosts the toxin may be required for virulence. However, it cannot be excluded that, in some hosts, AIP56 may have a less important contribution for virulence and that other virulence determinants, so far unidentified, may assume a more prominent virulence role. In this context, the in-depth characterization of the pathogenic role played by AIP56 in different *Phdp* hosts is of outmost importance to gain a more complete understanding of *Phdp* pathogenicity. So far, that characterization has been hampered by the lack of isogenic mutants deficient in *aip56* due to technical drawbacks mainly related to the fact that the toxin is encoded in a high copy plasmid.

In conclusion, the results reported in this work disclose the existence of differences in the susceptibility to AIP56 between sea bass and sea bream and emphasize the need for additional studies, to not only determine the susceptibility to this toxin and its role for virulence in other *Phdp* susceptible hosts, but also the contribution of other virulence factors for *Phdp*-induced lethality. Uncovering the relevance of the toxin in the context of the *Phdp* infections and the dissection of the specific effects triggered in different hosts is highly relevant for supporting future development of therapeutic and prophylactic measures against *Phdp* infections.

## 4. Materials and Methods

### 4.1. Production and Fluorescence Labeling of Recombinant Proteins

Full-length His-tagged AIP56 (AIP56) or catalytically inactive AIP56 (AIP56^AAIVAA^) were expressed and purified as previously described [16]. Briefly, the sequence encoding AIP56 from the MT1415 strain was cloned in pET28a(+) in frame with a C-terminal His-tag and expressed in *E. coli* BL21 (DE3) cells at 17 °C. Induced bacterial cells were lysed by sonication, centrifuged, and the recombinant protein was purified from the supernatant using nickel-affinity chromatography (Ni-NTA agarose, ABT, Madrid, Spain) followed by size exclusion chromatography (Superose 12 10/300 GL, GE Healthcare, Freiburg, Germany). AIP56^AAIVAA^ was purified from inclusion bodies by metal-affinity chromatography under denaturing conditions, refolded, and subjected to size exclusion chromatography [16]. Purified protein batches were analyzed by SDS-PAGE and purities, determined by densitometry of Coomassie blue-stained gels, were ≥ 90%. Recombinant AIP56 labelled with Alexa Fluor 488 (AIP56-488) was prepared using the Molecular Probes^®^ Alexa Fluor protein labelling kit (Thermo Fisher Scientific, Waltham, MA, USA), following the manufacturer’s instructions.

### 4.2. Determination of Recombinant Protein Concentration

Protein concentrations were determined by spectrophotometry by measuring the absorbance at 280 nm using NanoDrop 1000 (Thermo Fisher Scientific, Waltham, MA, USA), considering the extinction coefficients and the molecular weights, calculated by the ProtParam tool [31]

### 4.3. Fish

Four different lots of sea bream (*S. aurata*) and two lots of sea bass (*D. labrax*) with no previous history of *Phdp* infection were purchased from commercial hatcheries, and maintained in 600 L seawater aquaria. Water temperature was maintained at 20 ± 2 °C, salinity at 23–28‰, and the photoperiod was 14 h light: 10 h dark. Water quality was maintained with mechanical and biological filtration and ozone-disinfection and the fish were fed on commercial pellets (Skretting), adjusting the food intake to fish species/size and water temperature, according to the supplier’s recommendations. All experimental protocols were carried out in accordance with European and Portuguese legislation for the use of animals for scientific purposes (Directive 2010/63/EU; Decreto-Lei 113/2013) and were licensed by the DGAV (Lic. 0421/000/000/2021).

### 4.4. Cells

Peritoneal leukocytes were obtained from resting or inflamed peritoneal cavities of sea bream (weighting between 6.6 g and 173 g) or sea bass (weighting between 43 g and 367 g), following a procedure described in detail elsewhere [23,32]. Inflammation was induced by intraperitoneal (i.p.) injection of 100 µL of Incomplete Freund’s Adjuvant (IFA) 12 days before collecting the cells [33]. Cells were used at a density of 2 × 10^6^ cells/mL in L-15 medium (Gibco) adjusted to 322 mOsm and supplemented with 2.5% fetal bovine serum (FBS, Gibco), 1% penicillin/streptomycin (P/S, Gibco) and 20 U/mL heparin (Braun).

### 4.5. Cell Intoxication Assays

Intoxication assays were performed by incubating peritoneal leukocytes suspensions from sea bream (Batch IV, 65–173 g) or sea bass (Batch II, 202–367 g) with 10 µg/mL of AIP56 for 5 h at 22 °C. Untreated cells were used as controls. Occurrence of apoptosis and other signs of toxicity was assessed as described [20], through morphological analysis by light microscopy of at least 300 cells in cytospin preparations stained with Hemacolor (Merck), after labelling the neutrophils’ peroxidase using the Antonow’s technique [23,32].

### 4.6. AIP56 Internalization Assays

To assess AIP56 internalization, peritoneal leukocyte suspensions collected from sea bream (Batch II, 6.6–14.8 g and Batch IV, 75–158 g) or sea bass (Batch I, 43–65 g and Batch II, 202–367 g) were incubated with 5 µg/mL AIP56-488 in L-15 medium adjusted to 322 mOsm and supplemented with 2.5% FBS, 1% P/S and 20 U/mL heparin for 15 min on ice. Cell suspensions were centrifuged (5 min, 500 g, 4 °C), the supernatant discarded, and the cells resuspended in the same medium without toxin and incubated for 15 min at 22 °C. Cells were washed thrice with sea bass PBS (sbPBS; 50 mM phosphate buffer pH 7.2 + 184 mM NaCl), resuspended in 0.2 mL FACS buffer (sbPBS + 2% FBS), filtered through a 35 µm nylon mesh cell strainer into flow cytometry tubes and analyzed by flow cytometry on a BD FACSCanto™ II (BD Biosciences, San Jose, CA, USA). PI (5 µg/mL) was added to access cell viability and fluorescence were collected through a 530/30 nm (FITC) and 695/40 nm (PerCP-Cy5.5) detection filters. Ten thousand events per sample were analyzed at medium flow rate. All data were analyzed using FlowJo™ software v10.6.1 (San Diego, CA, USA).

### 4.7. Detection of aip56 in Phdp Field Isolates

The presence of *aip56* was evaluated in 49 isolates recovered during *Phdp* outbreaks occurred in Europe between 2014 and 2020 (Table 1). Strains ATCC29690 and EPOY8803-II, both lacking *aip56* [1,18], were used as negative controls, and strain MT1415 was used as a positive control. Bacterial stocks that were kept at −80 °C were plated in Tryptic Soy Agar (TSA, Difco) supplemented with NaCl to a final concentration of 1% (TSA-1) and grown for 48 h at 25 °C. A single colony of each strain was picked, with a sterile tip, and dipped into NZYTaq II 2x Green Master Mix (NZYTech). Colony-PCR was carried out with primers: 5′-GCATGACAGCAATATTTTCT-3′ (forward) and 5′-TTAATTAATGAATTGTGGCG-3′(reverse) and the following program: (a) denaturation for 3 min at 95 °C, (b) 35 cycles of denaturation at 95 °C for 30 s, annealing at 55 °C for 30 s, elongation at 72 °C for 1 min and (c) a final elongation step at 72 °C for 10 min. PCR products were analyzed by electrophoresis in an 1% agarose gel.

### 4.8. Comparative Analysis of AIP56 from Different Phdp Strains

Nucleotide sequences of *aip56* genes from *Phdp* strains isolated from sea bass (GRDL19-1, and SPDL20-5) and sea bream (DI21, GRSA19-1, SPSA20-2) were determined by Sanger sequencing at Eurofins Genomics or GATC Company. AIP56 ORFs were amplified by PCR using NZYProof DNA polymerase (NZYTech) and primers 5′-GCGCCATGGCGGTATATTAACAGGATGTCACAGTG-3′ (forward) and 5′- CGGCTCGAGCGGTTCTACTTCTACAAAATTTTGCGGC-3′ (reverse) and cloned into pET28a(+) using the NcoI and XhoI restriction sites. The sequence of *aip56* from MT1415 was obtained from GenBank (Accession number: TJZ88254.1). The DNA sequences were translated into protein using ExPASy translate tool [34] on the basis of standard genetic code. Multiple alignments of nucleotide/amino acid sequences were performed by Clustal Omega program [35].

### 4.9. Cloning Sea Bream NF-kB p65

To amplify and clone sea bream NF-kB p65 coding sequence, a sea bream was stimulated i.p. with 150 µL of a suspension of UV-killed *Phdp* MT1415 cells (OD600_nm_ = 0.9) and total RNA was extracted from the head-kidney using TripleXtractor reagent (Grisp) according to the manufacturer’s instructions. Briefly, 100 mg of the tissue were homogenized with TripleXtractor, chloroform was added to the homogenate and the aqueous phase collected. Following precipitation with isopropanol, the total RNA was washed with 70% ethanol and dissolved in RNAse-free water. cDNA synthesis was carried out using the PrimeScript RT Reagent Kit (Takara) according to the manufacturer’s instructions. The cDNA was then amplified using NZYProof DNA Polymerase (NZYTech) and primers 5′-GCGCCATGGAGGGTGGGTTTGGATGGAGCC-3′ (forward) and 5′-GCGCTCGAGAGTTGGGTGTCCTGACACAAACC-3′ (reverse) and cloned into pET28a(+) using the NcoI and XhoI restriction sites.

### 4.10. NF-kB Cleavage Assay

Sea bass and sea bream ^35^S-labeled NF-kB p65 synthetized in vitro using the TNT T7 Quick Coupled Transcription/Translation system (Promega), as previously described [16], were incubated in 10 mM Tris-HCl pH 8.0 + 300 mM NaCl for 2 h at 20 °C with different concentrations of AIP56 (0.5, 1, 10 and 100 nM) or with 100 nM of AIP56^AAIVAA^. Samples were subjected to SDS-PAGE, transferred to nitrocellulose membranes (Amersham Protran), and p65 cleavage analyzed by autoradiography.

### 4.11. In Vivo Toxicity Tests

The in vivo toxicity of AIP56 was tested in 6 independent experiments, using 4 different batches of seabream and 2 batches of sea bass, as specified in Table 3. In each experiment, groups of sea bream were injected with the indicated concentrations of AIP56, the catalytically inactive AIP56^AAIVAA^ or vehicle, in 100 µL of sbPBS. Sea bass, which are known to be susceptible to AIP56 [8], were injected in parallel. Injections were performed under anesthesia induced by immersion in 0.03% (*v*/*v*) ethylene glycol monophenyl ether. Injected fish were monitored at least 4 times a day and mortalities recorded. Any fish showing signs of advanced disease (darkening of body color, lethargy, erratic swimming and loss of equilibrium) were euthanized, by immersion in 0.06% (*v*/*v*) of ethylene glycol monophenyl ether followed by sectioning of the ventral aorta, and counted as dead.

In experiments 3 and 4, in parallel to the groups used for assessing lethality, groups were included for collecting samples for histological analysis. Two sampling times were established, at 36 and 60 h post-injection, for collecting the head-kidney, spleen, and liver. Five animals from each group were euthanized and sampled at 36 h and the remaining fish were sampled at 60 h. In experiment 3, two sea bass injected with AIP56 were found dead at 35 and 59 h after injection and three sea bass from the same group become moribund at 14, 18 and 24 h. These moribund fish were euthanized and organs collected. Tissues were fixed in 10% buffered formalin, routinely processed in an automated system, embedded in paraffin, sectioned at 3 µm, and stained with hematoxylin and eosin (H&E).

### 4.12. GenBank Accession Numbers

The nucleotide sequences of *aip56* from strains DI21, GRSA19-1, GRDL19-1, SPSA20-2, and SPDL20-5 were deposited in the GenBank database under accession numbers OM056532, OM056531, OM056529, OM056530, OM056528, respectively.

## Figures and Tables

**Figure 1 toxins-14-00119-f001:**
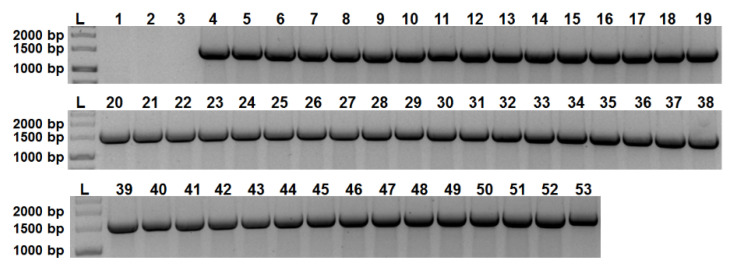
*aip56* is widespread amongst *Phdp* field isolates recovered from diseased sea bream and sea bass. Colony-PCR for *aip56* gene. L, DNA ladder marker (GeneRuler 1 kb DNA Ladder, Thermo Scientific™); 1, EPOY8803-II strain (negative control); 2, ATCC29690 strain (negative control); 3, no template (negative control); 4, MT1415 strain (positive control); 5–53, field isolates (see Table 1).

**Figure 2 toxins-14-00119-f002:**
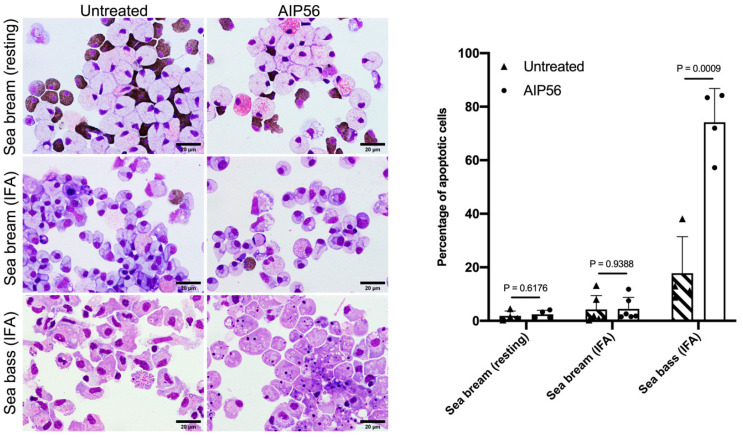
Sea bream leukocytes are resistant to AIP56 intoxication. Peritoneal leukocytes collected from resting or IFA-stimulated sea bream (Batch IV, 65–173 g) or from IFA-stimulated sea bass (Batch II, 202–367 g) were left untreated or incubated for 5 h at 22 °C with 10 µg/mL AIP56. Images show representative microscopic fields of cytospin preparations stained with Hemacolor after labeling the neutrophils’ peroxidase (brown). Note that after incubation with the toxin, most sea bass leukocytes were apoptotic, whereas no signs of toxicity were detected in sea bream leukocytes. The percentage of apoptotic leukocytes, determined by morphological analysis of at least 300 cells, is depicted in the graph. Results are expressed as mean ± SD (*n* = 4 for sea bream resting, *n* = 6 for sea bream IFA-stimulated, *n* = 4 for sea bass IFA-stimulated). Statistical analysis involved performing two-tailed Student’s *t*-test.

**Figure 3 toxins-14-00119-f003:**
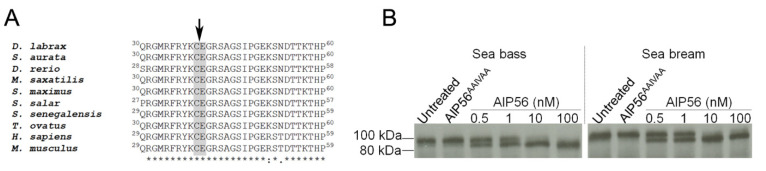
Sea bream NF-kB p65 is susceptible to AIP56 cleavage. (**A**) Alignment of the p65 N-terminal region from different species. *Dicentrarchus labrax* sequence (DLAgn_00141590) was retrieved from [21] and the remaining sequences were obtained from GenBank (Accession numbers: *Sparus aurata*, XP_030285950; *Danio rerio*, NP_001001839; *Morone saxatilis*, XP_035530725; *Scophthalmus maximus*, XP_035483131; *Salmo salar*, XP_014062259; *Solea senegalensis*, XP_043876337; *Trachinotus ovatus*, QJQ40092; *Homo sapiens*, XP_011543508; *Mus musculus*, NP_033071). AIP56 cleavage site (Cys/Glu) is indicated with an arrow and shaded in grey. Alignment was performed using Clustal Omega [22]. (**B**) ^35^S-labeled NF-kB p65 from sea bream or sea bass (positive control) were left untreated or incubated for 2 h at 20 °C with 0.5, 1, 10, or 100 nM AIP56 or with 100 nM of the catalytically inactive AIP56^AAIVAA^ and cleavage of p65 assessed by autoradiography. Result shown is representative of 3 independent experiments.

**Figure 4 toxins-14-00119-f004:**
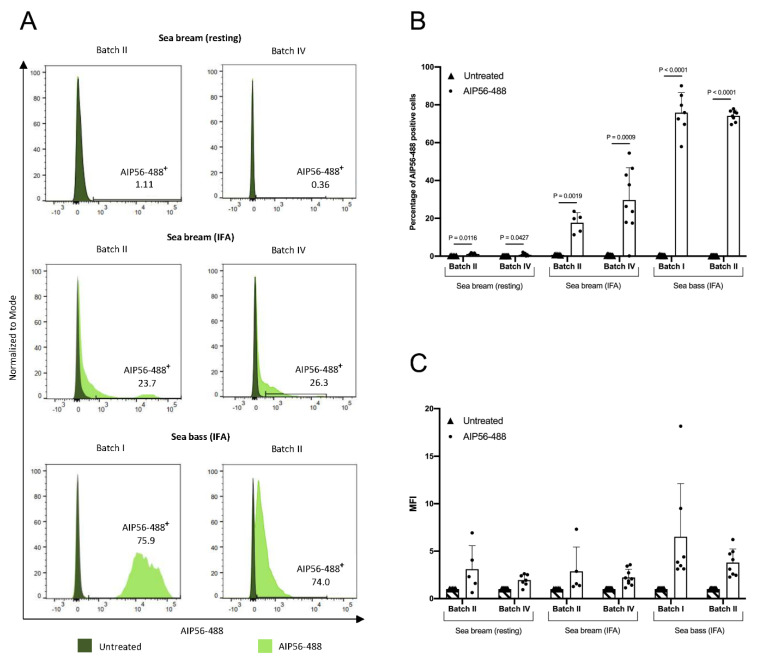
Internalization of AIP56 by sea bream peritoneal leukocytes. Peritoneal leukocytes collected from resting or IFA-stimulated sea bream, or from IFA-stimulated sea bass (positive control) were left untreated or incubated with 5 µg/mL AIP56-488 (15 min on ice + 15 min at 22 °C) and the percentage of AIP56-488-positive cells quantified by flow cytometry. (**A**) Representative flow cytometry plots. (**B**) Quantification of the percentage of AIP56-488-positive cells (*n* = 7 for resting sea bream/Batch II, *n* = 9 for IFA-stimulated sea bream/Batch II, *n* = 5 for resting and IFA-stimulated sea bream/Batch IV, *n* = 8 for sea bass/Batch I, *n* = 7 for sea bass/Batch II). Statistical analysis involved performing two-tailed Student’s *t*-test. (**C**) Mean fluorescence intensity (MFI) of AIP56-488 in the AIP56-488-positive cells, relative to untreated cells. Results are expressed as mean ± SD.

**Figure 5 toxins-14-00119-f005:**
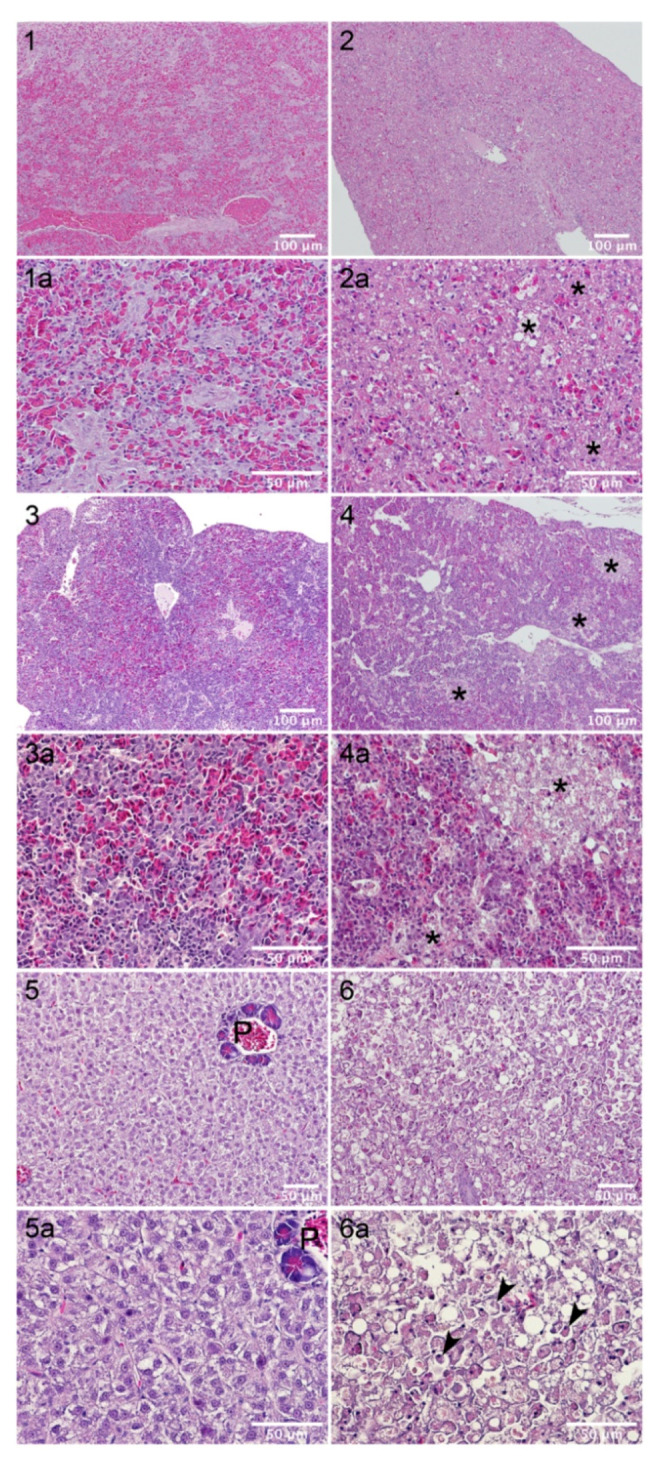
Histopathological changes induced by AIP56 in sea bass. (**1**) Spleen from a control fish (injected with vehicle); (**2**) Spleen collected 36 h after injection of AIP56 (0.7 μg AIP56/g body weight); (**3**) Head-kidney from a control fish (injected with vehicle); (**4**) Head-kidney collected 36 h after injection of AIP56 (0.7 μg AIP56/g body weight); (**5**) Liver from a control fish (injected with vehicle); (**6**) Liver collected 36 h after injection of AIP56 (0.7 μg AIP56/g body weight). Each panel labeled with “a” corresponds to a higher magnification of the panel with the respective number. Asterisks indicate necrotic foci and arrows indicate pyknotic/degenerated cells. P—pancreatic tissue.

**Figure 6 toxins-14-00119-f006:**
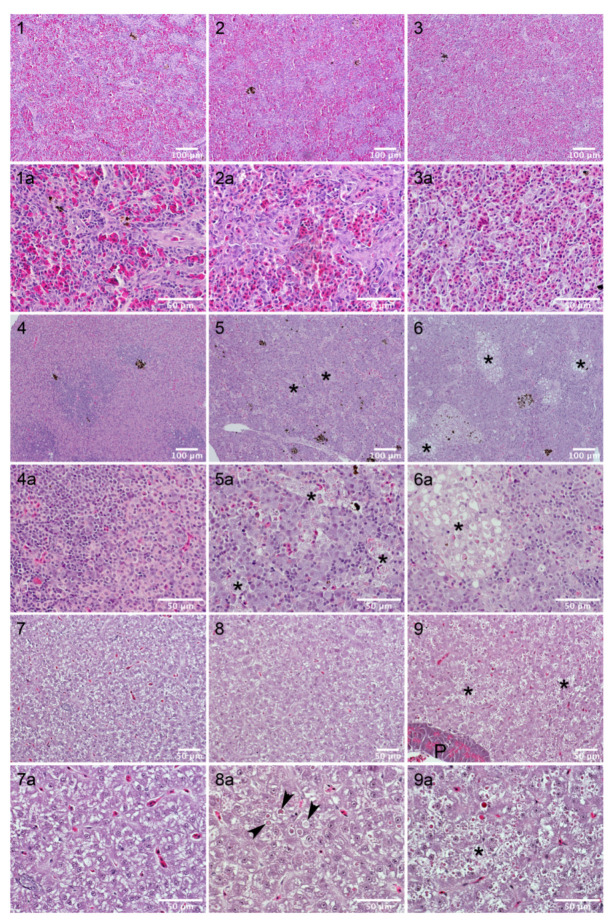
Histopathological changes induced by AIP56 in sea bream. (**1**) Spleen from a control fish (injected with vehicle); (**2**) Spleen collected 60 h after injection of AIP56 (Experiment 3, 3.1 μg AIP56/g body weight); (**3**) Spleen collected 30 h after injection of AIP56 (Experiment 4, 2.2 μg AIP56/g body weight) (**4**) Head-kidney from a control fish (injected with vehicle); (**5**,**6**) Head-kidneys collected 60 h after injection of AIP56 (Experiment 4, 2.2 μg AIP56/g body weight); (**7**) Liver from a control fish (injected with vehicle); (**8**,**9**) Livers collected 60 h after injection of AIP56 (Experiment 4, 2.2 μg AIP56/g body weight). Each panel labeled with “a” corresponds to a higher magnification of the panel labeled with the respective number. Asterisks indicate necrotic foci and arrows indicate necrotic cells. P—pancreatic tissue.

**Table 1 toxins-14-00119-t001:** *Photobacterium damselae* subsp. *piscicida* strains used in this study.

	Strain	Host Species	Year	Geographical Origin
1	ATCC29690	*Seriola quinqueradiata*	1972	Japan
2	EPOY8803-II	*Epinephelus akaara*	1988	Japan
3	DI21	*Sparus aurata*	1990	Spain
4	MT1415	*Dicentrarchus labrax*	unknown	Italy
5	SPSA14-1	*Sparus aurata*	2014	Spain
6	SPDL15-1	*Dicentrarchus labrax*	2015	Spain
7	SPSA16-1	*Sparus aurata*	2016	Spain (Mediterranean)
8	SPDL17-1	*Dicentrarchus labrax*	2017	Spain (Atlantic)
9	SPDL17-3	*Dicentrarchus labrax*	2017	Spain (Atlantic)
10	SPDL17-5	*Dicentrarchus labrax*	2017	Spain (Atlantic)
11	SPDL17-2	*Dicentrarchus labrax*	2017	Spain (Canary Islands)
12	SPDL17-4	*Dicentrarchus labrax*	2017	Spain (Mediterranean)
13	GRDL17-1	*Dicentrarchus labrax*	2017	Greece
14	SPSA18-1	*Sparus aurata*	2018	Spain (Canary Islands)
15	SPSA18-2	*Sparus aurata*	2018	Spain (Mediterranean)
16	SPDL18-1	*Dicentrarchus labrax*	2018	Spain (Atlantic)
17	SPDL18-2	*Dicentrarchus labrax*	2018	Spain (Mediterranean)
18	SPDL18-3	*Dicentrarchus labrax*	2018	Spain (Mediterranean)
19	SPDL18-4	*Dicentrarchus labrax*	2018	Spain (Mediterranean)
20	SPDL18-5	*Dicentrarchus labrax*	2018	Spain (Mediterranean)
21	SPDL18-6	*Dicentrarchus labrax*	2018	Spain (Mediterranean)
22	SPDL18-7	*Dicentrarchus labrax*	2018	Spain (Mediterranean)
23	ITDL18-1	*Dicentrarchus labrax*	2018	Italy
24	SPSA19-1	*Sparus aurata*	2019	Spain
25	SPSA19-2	*Sparus aurata*	2019	Spain (Canary Islands)
26	GRSA19-1	*Sparus aurata*	2019	Greece
27	SPDL19-2	*Dicentrarchus labrax*	2019	Spain
28	SPDL19-3	*Dicentrarchus labrax*	2019	Spain (Atlantic)
29	SPDL19-4	*Dicentrarchus labrax*	2019	Spain (Canary Islands)
30	SPDL19-1	*Dicentrarchus labrax*	2019	Spain (Mediterranean)
31	SPDL19-5	*Dicentrarchus labrax*	2019	Spain (Mediterranean)
32	SPDL19-6	*Dicentrarchus labrax*	2019	Spain (Mediterranean)
33	SPDL19-7	*Dicentrarchus labrax*	2019	Spain (Mediterranean)
34	SPDL19-8	*Dicentrarchus labrax*	2019	Spain (Mediterranean)
35	SPDL19-9	*Dicentrarchus labrax*	2019	Spain (Mediterranean)
36	SPDL19-10	*Dicentrarchus labrax*	2019	Spain (Mediterranean)
37	GRDL19-1	*Dicentrarchus labrax*	2019	Greece
38	GRDL19-2	*Dicentrarchus labrax*	2019	Greece
39	ITDL19-1	*Dicentrarchus labrax*	2019	Italy
40	SPSA20-1	*Sparus aurata*	2020	Spain (Mediterranean)
41	SPSA20-2	*Sparus aurata*	2020	Spain (Mediterranean)
42	GRSA20-1	*Sparus aurata*	2020	Greece
43	SPDL20-1	*Dicentrarchus labrax*	2020	Spain (Atlantic)
44	SPDL20-2	*Dicentrarchus labrax*	2020	Spain (Atlantic)
45	SPDL20-5	*Dicentrarchus labrax*	2020	Spain (Atlantic)
46	SPDL20-6	*Dicentrarchus labrax*	2020	Spain (Atlantic)
47	SPDL20-3	*Dicentrarchus labrax*	2020	Spain (Mediterranean)
48	SPDL20-4	*Dicentrarchus labrax*	2020	Spain (Mediterranean)
49	SPDL20-7	*Dicentrarchus labrax*	2020	Spain (Mediterranean)
50	SPDL20-8	*Dicentrarchus labrax*	2020	Spain (Mediterranean)
51	GRDL20-1	*Dicentrarchus labrax*	2020	Greece
52	GRDL20-2	*Dicentrarchus labrax*	2020	Greece
53	PTDL20-1	*Dicentrarchus labrax*	2020	Portugal

**Table 2 toxins-14-00119-t002:** Results of the in vivo AIP56 lethality tests.

Experiment	Species	Average Weight (g)	Batch	Treatment	Dose (µg/g Body Weight)	Mortality
1	*Sparus aurata*	11.4 ± 2.6	I	AIP56	0.9	0% (0/10)
				AIP56	0.09	0% (0/10)
	*Dicentrarchus labrax*	9.6 ± 2.1	I	AIP56	1	100% (10/10)
				AIP56	0.1	60% (6/10)
2	*Sparus aurata*	12.0 ± 2.0	I	AIP56	4.2	0% (0/10)
				AIP56	0.8	0% (0/10)
	*Dicentrarchus labrax*	10.4 ± 2.1	I	AIP56	1	100% (10/10)
				AIP56	0.1	90% (9/10)
3	*Sparus aurata*	16.0 ± 3.8	I	AIP56	3.1	0% (0/10)
	*Dicentrarchus labrax*	14.8 ± 6.6	I	AIP56	0.7	100% (10/10)
				AIP56	0.07	70% (7/10)
				Vehicle	x	0% (0/10)
4	*Sparus aurata*	11.6 ± 2.8	II	AIP56	2.2	100% (10/10)
				AIP56	0.2	30% (3/10)
				AIP56^AAIVAA^	2.2	0% (0/10)
				Vehicle	x	0% (0/10)
	*Dicentrarchus labrax*	53.6 ± 12.5	I	AIP56	0.2	90% (9/10)
5	*Sparus aurata*	34.4 ± 4.5	III	AIP56	2.9	0% (0/6)
	*Dicentrarchus labrax*	177.8 ± 23.2	II	AIP56	0.06	100% (5/5)
6	*Sparus aurata*	53.9 ± 14.5	IV	AIP56	1.9	0% (0/5)
	*Dicentrarchus labrax*	183.4 ± 41.0	II	AIP56	0.005	80% (4/5)

**Table 3 toxins-14-00119-t003:** Description of the in vivo assays.

Experiment	Species	Average Weight (g)	Batch	Treatment	Dose (µg/g Body Weight)	Lethality	Histopathology
1	*Sparus aurata*	11.4 ± 2.6	I	AIP56	0.9	Yes (*n* = 10)	No
				AIP56	0.09	Yes (*n* = 10)	No
	*Dicentrarchus labrax*	9.6 ± 2.1	I	AIP56	1	Yes (*n* = 10)	No
				AIP56	0.1	Yes (*n* = 10)	No
2	*Sparus aurata*	12.0 ± 2.0	I	AIP56	4.2	Yes (*n* = 10)	No
				AIP56	0.8	Yes (*n* = 10)	No
	*Dicentrarchus labrax*	10.4 ± 2.1	I	AIP56	1	Yes (*n* = 10)	No
				AIP56	0.1	Yes (*n* = 10)	No
3	*Sparus aurata*	16.0 ± 3.8	I	AIP56	3.1	Yes (*n* = 10)	Yes (*n* = 10)
				Vehicle	x	No	Yes (*n* = 10)
	*Dicentrarchus labrax*	14.8 ± 6.6	I	AIP56	0.7	Yes (*n* = 10)	Yes (*n* = 10)
				AIP56	0.07	Yes (*n* = 10)	No
				Vehicle	x	Yes (*n* = 10)	Yes (*n* = 10)
4	*Sparus aurata*	11.6 ± 2.8	II	AIP56	2.2	Yes (*n* = 10)	Yes (*n* = 10)
				AIP56	0.2	Yes (*n* = 10)	No
				AIP56^AAIVAA^	2.2	Yes (*n* = 10)	No
				Vehicle	x	Yes (*n* = 10)	Yes (*n* = 10)
	*Dicentrarchus labrax*	53.6 ± 12.5	I	AIP56	0.2	Yes (*n* = 10)	No
5	*Sparus aurata*	34.4 ± 4.5	III	AIP56	2.9	Yes (*n* = 6)	No
	*Dicentrarchus labrax*	177.8 ± 23.2	II	AIP56	0.06	Yes (*n* = 5)	No
6	*Sparus aurata*	53.9 ± 14.5	IV	AIP56	1.9	Yes (*n* = 5)	No
	*Dicentrarchus labrax*	183.4 ± 41.0	II	AIP56	0.005	Yes (*n* = 5)	No

## Data Availability

All data are reported in the manuscript.

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
