# Peer review of "Susceptibility of Sea Bream (Sparus aurata) to AIP56, an AB-Type Toxin Secreted by Photobacterium damselae subsp. piscicida"

_toxins, 2022, doi:10.3390/toxins14020119_

Round 1

Reviewer 1 Report

In this paper, Susceptibility of sea bream to AIP56 secreted by Photobacterium damselae subsp. Piscicida (Phdp) was studied in detail resulting in very meaningful results through the in vitro and in vivo assays demonstrating that sea bream was more resistant to AIP56 than sea bass was based on convincing evidence and concluding that the toxin plays a less toxic role in Phdp infection in sea bream. Mechanistically, it was elucidated that this resistance is related to the inefficient internalization of the toxin by the phagocytes. This is a somewhat unexpected finding and this manuscript is very well written and readable, so I don't have suggestions for revision.

However this conclusion that the toxin plays a less toxic role in Phdp infection in sea bream needs more solid evidence. It is also important to exclude that the sea bream used for the experiment had a history of Phdp infection. Are there differences in the genes encoding AIP56 in Phdp isolated from sea bass and sea bream? In addition, it must be excluded that the inefficient internalization of the toxin in sea bream is due to the tag protein attached to the expressed protein, in other word, the tag protein may prevent the binding of AIP56 to the receptor of phagocytes in sea bream. I also suggest that the author should point out the next research direction in the discussion section, that is, what further research should be carried out to fully prove the conclusions of this study, , such as the use of immunological technology to detect the absence of AIP56 in phagocytes of naturally infected sea bream. After all, there are structural differences between recombinant proteins and natural proteins.

Author Response

Point 1: In this paper, Susceptibility of sea bream to AIP56 secreted by Photobacterium damselae subsp. piscicida (Phdp) was studied in detail resulting in very meaningful results through the in vitro and in vivo assays demonstrating that sea bream was more resistant to AIP56 than sea bass was based on convincing evidence and concluding that the toxin plays a less toxic role in Phdp infection in sea bream. Mechanistically, it was elucidated that this resistance is related to the inefficient internalization of the toxin by the phagocytes. This is a somewhat unexpected finding and this manuscript is very well written and readable, so I don't have suggestions for revision. However this conclusion that the toxin plays a less toxic role in Phdp infection in sea bream needs more solid evidence. It is also important to exclude that the sea bream used for the experiment had a history of Phdp infection.

Authors’s response 1: The fish used in the studies reported in the manuscript were purchased when they had a small size (5-10g) and were maintained in the i3S aquarium systems until used in the experiments. None of the fish had previous history of Phdp infection. We added this information to the material and methods section “4.3 Fish”.

Point 2: Are there differences in the genes encoding AIP56 in Phdp isolated from sea bass and sea bream?

Authors’s response 2: This is indeed a very relevant question in the context of this work, that we have addressed, as described in the results’ section 2.1 and material and methods’ section 3.8. To assess if there were differences in the genes encoding AIP56 in Phdp isolated from sea bass and sea bream we compared the nucleotide sequences of the aip56 gene from 3 sea bream (DI21, GRSA19-1 and SPSA20-2) and 3 sea bass (MT1415, GRDL19-1 and SPDL20-5) strains isolated in different geographical locations. The results revealed that the aip56 genes from those strains are 100% identical, indicating that the strains secrete identical AIP56 toxins.

Despite this information was already included in the first version of the manuscript (in the last sentence of the results’ section 2.1), apparently it passed unnoticed for Reviewers 1 and 3. Therefore, to make this information clearer for the reader, in the revised manuscript we split section 2.1 into two sections (2.1 and 2.2). The numbering of the subsequent sections of the results were adjusted accordingly.

Point 3: In addition, it must be excluded that the inefficient internalization of the toxin in sea bream is due to the tag protein attached to the expressed protein, in other word, the tag protein may prevent the binding of AIP56 to the receptor of phagocytes in sea bream. I also suggest that the author should point out the next research direction in the discussion section, that is, what further research should be carried out to fully prove the conclusions of this study, such as the use of immunological technology to detect the absence of AIP56 in phagocytes of naturally infected sea bream. After all, there are structural differences between recombinant proteins and natural proteins.

Authors’s response 3: It is known that in some instances, the addition of certain tags to proteins may alter their function. AIP56 is an AB toxin that is able to intoxicate sea bass, mouse and human macrophages, through a mechanism that starts by the binding of its B subunit to a so far unknown cell surface receptor. The mechanism of AIP56 intoxication of sea bass and mammalian cells is similar and likely relies in the recognition of an evolutionarily conserved receptor (Silva, Pereira et al. 2013, Pereira, Pinto et al. 2014). Previous studies have shown that the addition of a C-terminal 6-histidine tag to AIP56 does not affect its toxicity for sea bass, mouse and human macrophages. Therefore, we think it is very unlikely that the tag would specifically affect binding to the receptor on sea bream cells

In response to this comment from reviewer 1, we made the following changes to the manuscript:

  1. The following sentence was added to the second paragraph of the Discussion: “For these studies, we used an AIP56 carrying a 6-histidine tag at the C-terminus, but is unlikely that this could explain the lack of toxicity of AIP56 for sea bream cells, because previous studies have shown that his-tagged AIP56 remains toxic for sea bass (Silva, Pereira et al. 2013, Pereira, Pinto et al. 2014), mouse [15] and human macrophages (unpublished results).”
  2. The following sentence was added at the end of the second paragraph of the Discussion: “Additionally, it would also be important to analyze (e.g by immunohistochemistry) the presence of AIP56 in phagocytes from sea bream naturally infected with Phdp, to confirm the here reported in vitro observations in a more biologically relevant scenario.”

References:

Pereira, L. M., R. D. Pinto, D. S. Silva, A. R. Moreira, C. Beitzinger, P. Oliveira, P. Sampaio, R. Benz, J. E. Azevedo, N. M. Dos Santos and A. do Vale (2014). "Intracellular trafficking of AIP56, an NF-kappaB cleaving toxin from Photobacterium damselae piscicida." Infect Immun 82(12): 5270-5285.

Silva, D. S., L. M. G. Pereira, A. R. Moreira, F. Ferreira-da-Silva, R. M. Brito, T. Q. Faria, I. Zornetta, C. Montecucco, P. Oliveira, J. E. Azevedo, P. J. B. Pereira, S. Macedo-Ribeiro, A. do Vale and N. M. S. dos Santos (2013). "The Apoptogenic Toxin AIP56 Is a Metalloprotease A-B Toxin that Cleaves NF-κb P65." PLoS Pathog 9(2): e1003128.

Reviewer 2 Report

This manuscript describes the difference in susceptibility against AIP56 between seabream and sea bass. At first, I wondered why the two species were selected to examine, but I understand that there are serial studies within this research field using them as described in the text.

Figures are better to be rewritten to add species names used in each subfigure to help readers to compare them.

Spell out “AB”; I though it as antibody.

Author Response

Point 1: This manuscript describes the difference in susceptibility against AIP56 between seabream and sea bass. At first, I wondered why the two species were selected to examine, but I understand that there are serial studies within this research field using them as described in the text.

Authors’s response 1: The studied species were selected because they are economically important species cultured in the Mediterranean area that are highly susceptible to Phdp infections and that, as the reviewer pointed out, have been used by different groups to study Phdp virulence mechanisms.

Point 2: Figures are better to be rewritten to add species names used in each subfigure to help readers to compare them.

Authors’s response 2: We re-checked the figures/figures’ legends and confirmed that the species names are indicated in every case. In Figures 2, 3 and 4, the species name is indicated in each picture and graphs contain a clear labeling that includes the species names. In Figures 5 and 6, the species names are not included in the pictures, because each figure contains data from only one species and the species name is mentioned in the first sentence of each figure legend: “Figure 5. Histopathological changes induced by AIP56 in sea bass.” and “Figure 6. Histopathological changes induced by AIP56 in sea bream.”

Point 3: Spell out “AB”; I though it as antibody.

Authors’s response 3: AB toxins are a group of bacterial toxins (that include diphtheria toxin, cholera toxin and anthrax toxin, amongst other) that are named AB because they possess two components: an enzymatic moiety (component A) that is responsible for the catalytic activity of the toxin, and a binding part (component B) that mediates binding of the toxin to the host cell membrane and delivery of the A subunit to its cytosolic destination. Considering that the “Toxins” journal is a specialized journal, we did not feel that it was necessary to define the meaning of “AB toxins” because the term is widely known by researchers working in the toxins’ field. Nevertheless, in response to this reviewer’s comment, to avoid misinterpretations, we made the following changes in the manuscript:

  1. The title “Susceptibility of sea bream (Sparus aurata) to AIP56, an AB toxin secreted by Photobacterium damselae subsp. piscicida” was changed to “Susceptibility of sea bream (Sparus aurata) to AIP56, an AB-type toxin secreted by Photobacterium damselae subsp. piscicida”
  2. In the Abstract, the sentence “The best characterized virulence factor of Phdp is the Apoptosis-Inducing Protein of 56 kDa (AIP56), a secreted AB toxin with metalloprotease activity against NF-kB p65 which has been shown to induce apoptosis of sea bass phagocytes during infection.” was changed to “The best characterized virulence factor of Phdp is the Apoptosis-Inducing Protein of 56 kDa (AIP56), a secreted AB-type toxin that has been shown to induce apoptosis of sea bass phagocytes during infection. AIP56 has an A subunit that displays metalloprotease activity against NF-kB p65 and a B subunit that mediates binding and internalization of the A subunit in susceptible cells.”

Reviewer 3 Report

Authors described an extensive study on best-characterized virulence factor of Phdp that is a key protein AIP56 responsible for causing toxicity in economically important fish species.

The manuscript has compared the virulence from different Phdp AIP56 proteins.

Here, it is crucial to mention/discuss/or present future remarks for the partnering proteins with AIP56 in different species and if any relevant explanation of the virulence for different fish species.

Providing aip56 gene and coding protein amongst Phdp field isolates recovered from diseased sea bream should be aligned to analyse if any functionally relevant deletion/insertion is present or not.

My last comment is why nanodrop is used for protein detection? This may cause low reliable outcomes due to its low resolution and high variability as well as incompatibility with detergents and denaturating agents in samples.

Author Response

Point 1: Authors described an extensive study on best-characterized virulence factor of Phdp that is a key protein AIP56 responsible for causing toxicity in economically important fish species. The manuscript has compared the virulence from different Phdp AIP56 proteins.Here, it is crucial to mention/discuss/or present future remarks for the partnering proteins with AIP56 in different species and if any relevant explanation of the virulence for different fish species.

Authors’s response 1: In the 4th paragraph of the Discussion, the section

“This result, which is in agreement with previous studies reporting a ubiquitous distribution of the aip56 gene in Phdp strains recovered worldwide from different fish species in the last 30 years [18,28,29], suggests the existence of selective pressure to maintain the toxin gene and supports the hypothesis that in the majority of hosts the toxin may be required for virulence. The in-depth characterization of the pathogenic role played by AIP56 in different Phdp hosts is of outmost importance to gain a more complete understanding of Phdp pathogenicity.” was changed to “This result, which is in agreement with previous studies reporting a ubiquitous distribution of the aip56 gene in Phdp strains recovered worldwide from different fish species in the last 30 years [18,28,29], suggests the existence of selective pressure to maintain the toxin gene and supports the hypothesis that in the majority of hosts the toxin may be required for virulence. However, it cannot be excluded that in some hosts AIP56 may have a less important contribution for virulence and that other virulence determinants, so far unidentified, may assume a more prominent virulence role. In this context, the in-depth characterization of the pathogenic role played by AIP56 in different Phdp hosts is of outmost importance to gain a more complete understanding of Phdp pathogenicity. "

Point 2: Providing aip56 gene and coding protein amongst Phdp field isolates recovered from diseased sea bream should be aligned to analyse if any functionally relevant deletion/insertion is present or not.

Authors’s response 2: When we first found that the sea bream cells were resistant to AIP56 toxicity (using AIP56 from the MT1415 strain, isolated from sea bass) we wondered if the resistance could be related to differences in the AIP56 sequence from sea bass and sea bream isolates. Therefore, as described in the results’ section 2.1 and material and methods’ section 3.8 (of the first version of the manuscript), we decided to compare the nucleotide sequences of the genes encoding AIP56 from 3 sea bream (DI21, GRSA19-1 and SPSA20-2) and 3 sea bass (MT1415, GRDL19-1 and SPDL20-5) isolates recovered from different geographical locations. The results revealed that the aip56 genes from those strains are 100% identical, indicating that all the strains analyzed secrete identical AIP56 toxins.

Despite this information was already included in the first version of the manuscript (in the last sentence of the results’ section 2.1), to make this information clearer for the reader, in the revised manuscript we split section 2.1 into two sections (2.1 and 2.2). The numbering of the subsequent sections of the results were adjusted accordingly.

Point 3: My last comment is why nanodrop is used for protein detection? This may cause low reliable outcomes due to its low resolution and high variability as well as incompatibility with detergents and denaturating agents in samples.

Authors’s response 3: Every method for protein concentration determination has pros and cons. We chose to use NanoDrop due to its simplicity, fast measurement time and robust performance and also because measurements can be done with microvolume samples. In NanoDrop, protein concentration is determined by measuring A280 and thus, the method can be applied to purified proteins that contain Tryptophan, Tyrosine, Phenylalanine residues or Cysteine-Cysteine disulphide bonds and exhibit absorbance at 280 nm. The A280 absorbance read is then combined with either the mass extinction coefficient or the molar extinction coefficient to calculate the concentration of the purified protein (Desjardins, Hansen et al. 2009). When executed properly, this is a reliable and reproducible method that can be used to determine the concentration of homogeneous solutions of purified proteins and that has been used in previous works to determine the concentration of AIP56 in solution (Silva, Pereira et al. 2013). We agree with the reviewer in that the weakness of the instrument is it incompatibility with detergents and denaturating agents in samples. However, the AIP56 solutions used in the studies reported in this manuscript do not contain any of these agents.

References used:

Desjardins, P., J. B. Hansen and M. Allen (2009). "Microvolume protein concentration determination using the NanoDrop 2000c spectrophotometer." Journal of visualized experiments : JoVE(33): 1610.